# Protein Hydroxylation by Hypoxia-Inducible Factor (HIF) Hydroxylases: Unique or Ubiquitous?

**DOI:** 10.3390/cells8050384

**Published:** 2019-04-26

**Authors:** Moritz J. Strowitzki, Eoin P. Cummins, Cormac T. Taylor

**Affiliations:** 1UCD Conway Institute, University College Dublin, Belfield, Dublin 4, Ireland; Moritz.strowitzki@ucd.ie (M.J.S.); eoin.cummins@ucd.ie (E.P.C.); 2School of Medicine, University College Dublin, Belfield, Dublin 4, Ireland; 3Systems Biology Ireland, University College Dublin, Belfield, Dublin 4, Ireland

**Keywords:** hypoxia, prolyl hydroxylation, hypoxia-inducible factor, mass spectrometry, HIF-prolyl hydroxylases, factor inhibiting HIF, Cep192, IKK-β, p105, p53, FOXO3a, OTUB1, MAPK6, RIPK4

## Abstract

All metazoans that utilize molecular oxygen (O_2_) for metabolic purposes have the capacity to adapt to hypoxia, the condition that arises when O_2_ demand exceeds supply. This is mediated through activation of the hypoxia-inducible factor (HIF) pathway. At physiological oxygen levels (normoxia), HIF-prolyl hydroxylases (PHDs) hydroxylate proline residues on HIF-α subunits leading to their destabilization by promoting ubiquitination by the von-Hippel Lindau (VHL) ubiquitin ligase and subsequent proteasomal degradation. HIF-α transactivation is also repressed in an O_2_-dependent way due to asparaginyl hydroxylation by the factor-inhibiting HIF (FIH). In hypoxia, the O_2_-dependent hydroxylation of HIF-α subunits by PHDs and FIH is reduced, resulting in HIF-α accumulation, dimerization with HIF-β and migration into the nucleus to induce an adaptive transcriptional response. Although HIFs are the canonical substrates for PHD- and FIH-mediated protein hydroxylation, increasing evidence indicates that these hydroxylases may also have alternative targets. In addition to PHD-conferred alterations in protein stability, there is now evidence that hydroxylation can affect protein activity and protein/protein interactions for alternative substrates. PHDs can be pharmacologically inhibited by a new class of drugs termed prolyl hydroxylase inhibitors which have recently been approved for the treatment of anemia associated with chronic kidney disease. The identification of alternative targets of HIF hydroxylases is important in order to fully elucidate the pharmacology of hydroxylase inhibitors (PHI). Despite significant technical advances, screening, detection and verification of alternative functional targets for PHDs and FIH remain challenging. In this review, we discuss recently proposed non-HIF targets for PHDs and FIH and provide an overview of the techniques used to identify these.

## 1. Hypoxia and the Hypoxia-Inducible Factor (HIF) Pathway

Most animals utilize O_2_ as a primary metabolic substrate during oxidative phosphorylation. Because of the absolute reliance on a continuous supply of O_2_, animals have evolved a mechanism to adapt to hypoxia at a cellular level [1]. This is mediated by the activation of hypoxia-inducible factors (HIFs). In hypoxia, where cellular oxygen demand exceeds supply, HIFs are stabilized and induce the transcription of HIF-target genes which promote an array of adaptive mechanisms including enhancement of anaerobic ATP generation through glycolysis [2], increased oxygen supply through angiogenesis and increased blood oxygenation through erythropoiesis [3]. In normoxia (where oxygen supply exceeds demand), HIFs are continuously degraded in an O_2_-dependent manner and, therefore, cannot induce the transcription of target genes, see Figure 1 [3,4].

HIFs consist of two subunits, termed HIF-α and -β (also called aryl hydrocarbon receptor nuclear translocator; ARNT). There are three HIF-α homologues (HIF-1α, HIF-2α, HIF-3α) which form dimers with HIF-1β [1,4,5]. HIF 1–3 bind to hypoxia-responsive elements (“HRE”) of HIF-target genes (genes that carry an HRE within their promotor region) to promote adaption to hypoxia [6]. Notably, recent studies revealed that HIFs can interact with other signaling pathways (including Notch [7], Wnt [8], and Myc [9] pathways) via different (non-HRE-mediated) mechanisms [4]. In addition, the functional significances of each of the HIF complexes (HIF 1, 2 and 3) while somewhat overlapping, are markedly different from each other [4]. Each HIF complex confers a distinct role during adaptation to hypoxia [4,5,10]. In summary, the HIF pathway is an ubiquitous cellular mechanism promoting transcriptional adaptation to hypoxia. 

## 2. HIF Hydroxylation

HIF-α is synthesized at a high basal rate; however, once synthesized, it undergoes rapid degradation provided sufficient non-mitochondrial oxygen is available. Oxygen-dependent targeting of HIF-α subunits to proteasomal degradation is conferred by the HIF prolyl 4-hydroxylases PHD1, PHD2 and PHD3 (also termed EGLN2, EGLN1 and EGLN3, respectively). In humans, PHDs hydroxylate HIF-α subunits on two proline residues (HIF-1α: Pro402 and 564; HIF-2α: Pro405 and Pro531) [11], thus triggering their recognition by a multimeric E3 ubiquitin ligase complex formed by the von Hippel-Lindau tumor suppressor protein (pVHL), elongin B and C, Cullin 2 and RING-box 1 proteins [12,13,14], see Figure 1. For hydroxylation to occur, PHDs rely on oxygen (O_2_), iron (Fe^2+^), α-ketoglutarate (also known as 2-oxaloglutarate; 2-OG) and ascorbic acid (vitamin C) [4]. PHDs use one oxygen atom from O_2_ during the oxidative decarboxylation of 2-OG, yielding succinate and CO_2_, and the other is incorporated directly into the oxidized amino acid residue of HIF-α, see Figure 2, upper panel [15]. PHDs are dioxygenases in that they incorporate both atoms of molecular oxygen into their targets. Without sufficient O_2_, 2-OG or Fe^2+^ present within the cell, PHDs are unable to hydroxylate HIF-α subunits to mark them for degradation. Therefore, in hypoxia, HIF-α subunits are stable and can bind to HIF-1β to form a transcriptionally active dimer which induces the transcription of HIF-target genes [4], see Figure 1. 

Factor-inhibiting HIF (FIH) is another enzyme which inhibits HIF-α signaling in normoxia by hydroxylating an asparagine residue (Asn803 in human HIF-1α; Asn851 in human HIF-2α) [16] within the C-terminal transactivation domain (CTAD) of HIF-1α and 2α, see Figure 2, lower panel. This hydroxylation prevents binding of the transcriptional co-activator histone acetyltransferase p300/CREB-binding protein (p300/CBP), thereby reducing the carboxy-terminal transactivation domain activity of HIF [17]. As with prolyl hydroxylation, asparaginyl hydroxylation by FIH is reduced in hypoxia, thereby de-repressing HIF transactivation. 

PHD enzymes display distinct and tissue-specific expression patterns. On a cellular level, PHD1 is predominantly expressed within the nucleus, whereas PHD2 can be found within the cytoplasm and PHD3 in both the nucleus and the cytoplasm [18,19,20]. Although all PHDs are expressed ubiquitously, they display tissue-specific differences in expression levels. PHD2, the most abundantly expressed PHD enzyme, can be found in almost every tissue, whereas within the testis and the heart, PHD1 and PHD3 are the most highly expressed enzymes, respectively [17]. In the context of HIF signaling, PHD2 is the primary oxygen sensor. Homozygous loss of the PHD2 gene is not compatible with life and a partial loss of PHD2 (PHD2^+/−^) or a cell-type specific loss of PHD2 strongly increases HIF expression [17]. In contrast, pre-clinical studies have shown that animals with a homozygous loss of PHD1 or 3 are viable. Further studies of transgenic animals have provided intriguing insights into the role of individual PHD enzymes during the development of chronic diseases. For example, PHD1-deficient mice are protected against colitis or liver pathologies such as biliary fibrosis or ischemia/reperfusion injury [21,22,23,24,25]. 

Of note, the PHD catalytic activity can be pharmacologically inhibited by PHD-inhibitors (PHIs), such as FG-4592 (Roxadustat) [26], the pan-hydroxylase inhibitor dimethyloxalylglycine (DMOG), or ethyl-3,4-dihydroxybenzoate (EDHB) [27]. The majority of the PHIs that are currently under investigation for clinical use are 2-oxoglutarate antagonists [18,27,28]. The fact that PHDs can be pharmacologically inhibited led to investigations into the potential of using PHIs for the treatment of pathologies, such as anemia, inflammation and ischemia [27,29,30]. The currently available PHIs are generally not enzyme-specific [18]. Therefore, understanding the full range of hydroxylase targets is important to fully elucidate the potential pharmacology of PHIs in patients.

## 3. Other Prolyl Hydroxylases

Post-translational modification through prolyl hydroxylation is a well-described phenomenon. In extra-cellular collagen, upwards of 30% of prolines may be hydroxylated which is conferred by a family of collagen prolyl hydroxylases (CPHs) localized within the endoplasmic reticulum [18,31]. CPHs consist of two different enzyme families, collagen prolyl 4-hydroxylases and collagen prolyl 3-hydroxylases, which have distinct functions [32]. Since prolyl 4-hydroxylation is, however, the single most prevalent post-translational modification in humans [33], we will focus on prolyl 4-hydroxylases within the present review. CPHs act on -X-Pro-Gly sequences to catalyze the formation of hydroxyproline in collagens and similar peptides, thus stabilizing the triple helix of collagen [18]. Both the formation of collagen and sufficiently stabilized collagen fibers within the connective tissue are important for wound healing but also fibrotic diseases defined by excessive collagen deposition. Thus, like PHDs, CPHs represent a potential target for the treatment of fibrotic diseases, such as liver fibrosis [34,35]. 

Like PHDs, CPHs use O_2_, Fe^2+^, α-ketoglutarate (2-OG) and ascorbic acid as co-substrates [36,37]. While PHDs specifically bind to the ODD of proteins such as HIF, CPHs have a collagen-specific binding domain. Studies of enzyme kinetics revealed that the Michaelis constant (Km) values of PHDs for O_2_ are higher than those of CPHs, which results in the capacity of PHDs to sense changes in cellular oxygen levels [38]. The Km values and thus the affinity for 2-OG and metal ions are likewise different between PHDs and CPHs [38,39]. As a matter of fact, non-specific pharmacological PHIs (for example, 2-OG mimetics or iron chelators) also have inhibitory effects on CPHs, thus making them potentially interesting therapeutics for fibrotic diseases [35,40,41,42]. Notably, most of the PHIs are more potent inhibitors of CPHs than of PHDs [38]. In this review, we will compare the methods and techniques used to screen, detect and verify targets of PHDs and FIH. For other comprehensive discussions of recently identified non-HIF targets of PHD- or FIH-mediated hydroxylation see recent reviews [12,17]. 

## 4. Methods Used to Detect Protein Hydroxylation 

The screening, detection and verification of alternative functional targets for PHD- or FIH-mediated post-translational modification is technically challenging. A cursory review of the literature reveals a large variance in the methods and techniques used to detect and confirm alternative targets for PHD- or FIH-mediated post-translational modification, see Table 1. In this review, we aim to provide an overview of the methods and techniques used to detect hydroxylation by PHDs and FIH, see Figure 3. In addition, we will discuss recent alternative PHD or FIH targets in terms of translational and physiological relevance and highlight the different techniques applied during the detection, screening and functional verification of PHD and FIH targets, see Table 1. 

### 4.1. Enzymatic or Kinetic Assays

One method to identify prolyl hydroxylation conferred by PHDs is to monitor their activity by measuring ^14^CO_2_ formed from [^14^C]-carboxyl-labeled substrates, such as radioactively labeled 2-OG [57,58,59]. These CO_2_ capture assays are based on the knowledge that hydroxylation by 2-OG-dependent dioxygenases results in the decarboxylation of 2-OG and the release of CO_2_. Hydroxylation in the presence of 2-OG radiolabeled with ^14^C at the carbon position leads to the release of radioactive CO_2_, which can then be captured with filters that are pre-saturated with calcium hydroxide (Ca(OH)_2_) [47,59]. Variations of this assay measure the consumption of oxygen using an O_2_-sensing electrode or quantify unreacted α-ketoglutarate by post-reaction derivatization that forms a fluorescent product [36,57,60]. Another enzymatic assay couples the formation of succinate to that of NAD+ formation via succinyl-coenzyme A synthetase, pyruvate kinase, and lactate dehydrogenase [61]. Alternatively, the enzyme activity can be analyzed by directly determining the amount of radioactive 4-hydroxyproline formed within the substrate [62,63]. One limitation that needs to be considered is that these kinetic assays are highly controlled in vitro studies, as purified enzymes and minimal co-substrates are typically used. Thus, the putative alternative targets that can be detected with this method are limited to those hydroxylated under the limited and, in many cases, non-physiologic conditions of the assay. For example, if a cellular scaffold protein is required to facilitate hydroxylation of a target protein, this will likely not be identified as a target in these assays. 

### 4.2. In Silico Screening

In silico screening of putative hydroxylation targets for the HIF-α “consensus sequence”, LXXLAP (with X referring to any amino acid), present in the oxygen-dependent degradation domain (ODD) of HIF-α subunits allows the identification of other putative substrates for PHD-mediated hydroxylation based on sequence similarities with known targets [12,64]. Potential alternative targets for PHD1 and 3 involved in the regulation of cell-cycle progression and glucose metabolism, respectively, have been detected in this way [53,65]. However, the strength of this “consensus motif” is unclear and it remains highly unlikely that all bona fide PHD-targets for hydroxylation will show the LXXLAP-motif or whether all LXXLAP-motif containing proteins are hydroxylated [12]. 

### 4.3. Mass Spectrometry

Mass spectrometry (MS) is one of the most frequently applied methods to detect protein hydroxylation in cellular extracts. In MS, the digested peptides are ionized and analyzed in terms of their specific masses and charges (mass/charge ratio) [66]. In general, hydroxylation of proline, lysine or asparagine involves the replacement of a hydrogen atom with a hydroxyl group, giving an overall mass increase of 16 Da [66]. However, the overall mass of hydroxyproline detected by MS (113Da) is the same as the overall mass of other unmodified amino acids, such as leucine and isoleucine. Therefore, mutations that change the amino acid sequence from proline to leucine can be mistaken for hydroxylation of proline residues [66]. Moreover, it is difficult to differentiate between spurious oxidation and enzymatic hydroxylation by MS since both modifications can occur in a variety of amino acid side chains and spurious oxidation likewise increases the mass by 16Da [12,47]. Therefore, mass spectrometry fragmentation data need to be of high resolution and high coverage to confirm the assignment for the putative modification that has been selected for further analysis by any screening method [12]. In the case of HIF, these setbacks could additionally be overcome by detecting and comparing the mass differences under specific conditions, such as hypoxia versus normoxia or using different pharmacological substrate-trapping strategies linked to MS or mutation of OH-acceptor residues [12,66]. 

The combination of MS and a pharmacological substrate-trapping strategy can facilitate the detection of, for example, transient FIH-mediated enzyme-substrate interactions [67]. To do this, enzyme-substrate interactions are stabilized by the pretreatment of cells with, for example, dimethyloxalylglycine (DMOG; a cell-penetrant inhibitor of 2OG-dependent oxygenases) [55,67].

### 4.4. Von Hippel-Lindau Tumor Suppressor (VHL) Capture Assay

Another way to detect putative prolyl hydroxylation targets is the von Hippel-Lindau protein (pVHL)-capture-based approach [31]. VHL, which normally detects prolyl hydroxylated HIF-α, can be genetically modified by different point mutations, thus enabling VHL to recognize a broader spectrum of hydroxyproline residues [31]. By using a set of different point-mutated VHL complexes in the pull-down of tryptic peptide digests, hydroxylated peptides can be detected [31]. Nonetheless, this method is restricted to VHL targets and other hydroxylated proteins may not be detected. Therefore, the capacity of this assay to identify new non-HIF hydroxylation targets is limited. A specific variation of this method has recently been applied to detect a new VHL substrate, zinc fingers and homeoboxes 2 (ZHX2; see specific section “ZHX2” below) [54]. 

### 4.5. Immunoprecipitation and Other Antibody-Based Techniques

Finally, the verification of hydroxylated peptides can be determined by a spectrum of different in vitro and in vivo immunodetection methods including advanced cloning techniques, immunoprecipitation, western blotting using for example specific anti-hydroxyproline (Pro402 and Pro564 [68]) and anti-hydroxyasparagine (Asn803 [69]) antibodies.

## 5. Evidence for HIF Hydroxylation

As outlined above, HIF-α subunits are the best-characterized targets for prolyl and asparaginyl hydroxylation. HIF target genes are involved in an array of cellular pathways that affect erythropoiesis, angiogenesis, cell differentiation and proliferation, cell metabolism and tumorigenesis [70,71]. In this context, HIF-α and HIF-β subunits have, in part, distinct functions to each other. During embryonic development, for example, mice lacking HIF-1β show a deleterious decrease in VEGF protein levels and thus are unable to survive since VEGF is essential for both angiogenesis and vasculogenesis [72,73,74]. Genetic loss of HIF-α subunits (HIF-1α or -2α), on the other hand, results in lethal vascular defects, cardiac malformation and bradycardia due to insufficient erythropoiesis and catecholamine production [71,75]. Here, we will focus on the hydroxylation of the HIF-α subunit, the best-characterized target for PHD- and FIH-mediated post-translational modifications [43,44,45,46,76]. 

In 2001, two research groups published papers reporting that, under normoxia, the HIF-1α subunit is targeted by PHDs and is rapidly destroyed via a mechanism that involves ubiquitylation by the von Hippel-Lindau tumor suppressor (pVHL) E3 ligase complex [43,44]. Using VHL capture assays and immunoprecipitation, the Ratcliffe group reported that this PHD-dependent modification is mediated by hydroxylation and that it occurs at the proline residue 564. In addition, PHDs were shown to absolutely rely on the co-factors oxygen (O_2_) and iron (Fe^2+^), thus, identifying PHDs as cellular oxygen sensors [43]. The Kaelin group applied MS to confirm hydroxylation of Pro564 (increase in molecular weight by 16Da) within the HIF-α subunit. Intriguingly, according to the experimental design of this study, the colleagues from the Kaelin group intentionally replaced methionine residues (Met561 and Met568) in close proximity to Pro564 with alanine prior to MS to prevent false positive results due to spurious oxidation of methionine residues [44]. Finally, a second hydroxylation site Pro402 was confirmed [45].

In 2002, Hewitson and colleagues first-described that the newly identified FIH [77] is responsible for previously detected regulatory hydroxylation of a specific asparagine residue (Asn803; [78]) within the C-terminal transactivation domain (CTAD) of HIF-1α [46]. After confirming that FIH is a HIF-α CTAD hydroxylase and that FIH does not hydroxylate HIF-α at the known target sites of PHDs, MS was used to confirm FIH-mediated hydroxylation within the CTAD of HIF-α [46]. Next, several mutations at the Asn803 residue were constructed and it was confirmed that FIH-conferred hydroxylation of Asn803 via 2-OG decarboxylation and interaction assays [46]. Further analysis, revealed that FIH activity depends on oxygen levels and can be directly inhibited by cobalt (II), which binds to the HIF-α subunit and thus prevents VHL binding [46]. 

Interestingly, there are significant differences between the oxygen-dependent regulation of each hydroxylation site in human HIF-1α. Using hydroxy residue-specific antibodies, it has been demonstrated that the hydroxylation of first, Pro402, then Pro564 and finally, Asn803 is suppressed with decreasing oxygen levels [68]. In addition, although each of the two hydroxylation sites can mediate VHL-conferred HIF-1α degradation, it seems evident that an efficient and more rapid protein degradation, one that is adequate to overcome the rate of synthesis, of HIF-1α requires hydroxylation of both Pro402 and Pro564 [79]. Surprisingly, clinical studies of tumor-derived tissue demonstrated only a weak correlation between the amount of HIF-1α expression and oxygen levels [80,81]. This might be explained by different tissue-specific hydroxylation patterns and by the fact that HIF-1α VHL-mediated degradation seems to be rate-limited and inhibited by hypoxia in tumors since many tumors with a positive HIF-1α expression showed a significant amount of prolyl-hydroxylated HIF-1α, even in the most hypoxic regions [82]. 

In conclusion, the detection and verification of PHD- and FIH-mediated HIF-α hydroxylation was first demonstrated using an array of techniques including MS, CO_2_ and VHL capture assays, and in silico methods in combination with substrate trapping strategies, which can serve as a model for the screening, detection and verification of other alternative hydroxylation targets. 

## 6. Evidence for Non-HIF Hydroxylation Targets 

### 6.1. NF-κB

Since the discovery of HIF, hundreds of genes have been reported to be sensitive to hypoxia. While HIF is a master-regulator of hypoxia-dependent gene expression, the transcriptional repertoire in response to low oxygen is not restricted to HIF, with multiple other transcription factors affected by hypoxic conditions and/or oxygen levels [83,84]. Of particular interest is the well-described link between hypoxia and regulation of the NF-κB pathway [85]. The hypoxic sensitivity of the NF-κB pathway was first described almost 25 years ago [86]. The sensitivity of the NF-κB pathway to hypoxia is at least, in part, conferred by the same HIF-prolyl hydroxylases. Here, we will summarize evidence relating to the regulation of IKKβ, a key kinase in the NF-κB pathway, by the PHDs. 

When investigating the molecular mechanisms underpinning the hypoxia-dependent regulation of NF-κB, it was observed that the pan-hydroxylase inhibitor, DMOG, promoted IkBα phosphorylation and enhanced NF-κB-luciferase reporter activity [48]. Using PHD-specific siRNAs, it was determined that PHD1 was the primary mediator of this. Intriguingly, both IKKα (P190) and IKKβ (P191) contain a conserved proline residue as part of an LXXLAP motif adjacent to key phosphorylation sites in the activation loop which is critical for the activation of the canonical NF-κB pathway (S180 IKKα and S181 IKKβ) [48]. This motif strongly resembles the oxygen-dependent degradation domains of HIF-α, through which HIF-α is hydroxylated and marked for interaction with the VHL E3-ubiquitin ligase. While HIF is marked for degradation via hydroxylation of the LXXLAP motif, the impact of hypoxia and/or hydroxylase inhibition on the NF-κB pathway is more modest. It should also be noted that direct proof of the hydroxylation of IKKβ by MS or CO_2_ release assay was not provided in this study [48].

Subsequent studies have supported alterations in hydroxylase-dependent signaling with changes in NF-κB activity [21,49,87,88,89,90,91]. The strongest direct evidence for hydroxylation of IKKβ was provided by Zheng and colleagues [47]. They employed a non-biased approach to identify novel PHD1 substrates from a breast cancer cDNA library. Their 96-well plate format in vitro decarboxylation assay identified hydroxylation signals for HIF-2α (a known hydroxylase substrate), as well as other substrates including IKKβ [47]. While IKKβ was not the main focus of this study, the identification of a hydroxylation signal on IKKβ by PHD1 using this non-biased approach adds further evidence in support of IKKβ being a target of functional hydroxylation.

While it was proposed that IKKβ may interact with VHL, in this case, hydroxylation does not appear to regulate protein stability [48]. Alternatively, hydroxylation of the proline residue on IKKβ may affect phosphorylation and thus activation of the kinase and, in this way, play an oxygen-sensitive modulatory role of the pathway. Recent evidence from Wang et al. supports this hypothesis [92]. This study described hypoxia-dependent inhibition of K63-ubiquitination of IKKβ. This impaired K63-ubiquitination of IKKβ was mirrored with the hydroxylase inhibitor DMOG. Furthermore, the authors observed enhanced K63-ubiquitination with overexpression of PHD1 that was associated with suppressed phosphorylation of IKKβ [92].

The hypothesis that P191 of IKKβ is important for the functional catalytic activity of the kinase is further supported by evidence that P191A mutants of IKKβ have impaired catalytic activity compared to wild-type controls [25]. As part of the same study, MS was used to identify prolyl hydroxylation of a synthetic IKKβ peptide containing the putative LXXLAP prolyl hydroxylation site. These peptide-based hydroxylation assays demonstrating prolyl hydroxylation of IKKβ in a rabbit reticulocyte lysate system were not, however, consistently observed in a lysate-free decarboxylation assay [25]. The reason for the apparent discrepancy between these two assay types is not immediately clear and may relate to specific criteria required for peptides to be hydroxylated in vitro. While hypoxia or pharmacologic hydroxylase activity increase basal NF-κB activity, the opposite is the case in cytokine-induced activity where hydroxylase inhibition decreases activated NF-κB signaling [48,89]. This underscores the complex relationship between both pathways. 

Thus, in summary, IKKβ is a candidate for prolyl hydroxylase-dependent modification that can affect the catalytic activity of the kinase and subsequent downstream signaling through the NF-κB pathway.

### 6.2. p53

The tumor suppressor gene, p53, is important in the regulation of a multitude of cellular responses including apoptosis, senescence, DNA repair, and cell cycle arrest [51,80,93,94,95]. Conflicting data exist relating to whether or not p53 protein expression is induced [96,97,98] or repressed [99,100] by hypoxia. Interestingly, recent work by Rodriguez and colleagues in which they showed the interaction of PHD3 and p53 may shed light on this controversy [51]. Using MS and pharmacological trapping strategies, this study reports that PHD3 hydroxylates p53 at proline 359, which regulates p53 protein stability through the modulation of ubiquitination [51]. Immunoprecipitation and western blotting confirmed the physical interaction of PHD3 and p53 [51]. 

However, it seems that this is only part of the story, as PHDs influence p53, localization, stability and activity on multiple levels. Within this context, it has been shown that PHD1 regulates p53 binding to p38α kinase in a hydroxylation-dependent manner which can cause resistance to chemotherapy in a xenograft colorectal cancer model [50]. Loss of PHD1 gene function attenuates p53 activation and reduces p53 phosphorylation at Ser15 which leads to an increased sensitivity of colorectal tumor cells to chemotherapy treatment in vivo and in vitro [50]. Notably, these effects appear to be HIF-independent, since HIF-1α or HIF-2α silencing did not affect the PHD1-mediated effects on p53 phosphorylation [50]. These data suggest that p53 is hydroxylated by PHD1; however, this study does not identify a specific hydroxylation site [50]. Previous screening assays did not identify p53 as a substrate of hydroxylation by PHD1 [47]. 

More recently, co-immunoprecipitation studies confirmed that PHD1 interacts with p53. In addition, tandem mass spectrometry (MS/MS) revealed that PHD1 most probably hydroxylates p53 at proline 142 [49]. Subsequent experiments using cells expressing a proline to alanine mutation at the residue 142 (Pro142Ala) within the mutated p53 gene were performed [49]. It could be confirmed that the p53 Pro142Ala mutation abolished the interaction between p53 and PHD1 [49]. Notably, Ullah and colleagues could show that PHD1 hydroxylation of p53 was HIF-dependent, as the knock-out of HIF-1α prevented PHD1-mediated p53 phosphorylation and hydroxylation [49]. In an animal model of acute skin inflammation, loss of PHD1 significantly reduced the overall inflammatory response while increasing cell death within the skin [49]. In contrast to the observed cell death due to loss of PHD1 in models of acute skin inflammation, findings in the intestine demonstrated that loss of PHD1 decreases epithelial cell apoptosis in a mouse model of chemically-induced colitis [21]. This illustrates that the role of PHD1 might well be tissue-specific regarding, for example, the inflammatory response and cell death. 

There is mounting evidence that both PHD1 and PHD3 can affect p53 protein stability and most probably function through the hydroxylation of proline 359 (PHD3) and 142 (PHD1). However, some controversy exists as to whether the subsequent effects of PHD1 on cellular chemotherapy resistance [50] or the inflammatory response [49] are HIF-independent. 

### 6.3. FOXO3a

The transcription factor forkhead box O (FOXO) is of relevance to diverse cancer regulatory pathways as it suppresses cell proliferation and survival by activating the expression of cancer-specific target genes [101,102,103]. Notably, recent genetic studies in mice and humans supported the role of forkhead proteins in cancer. In leukemia or prostate cancer, forkhead proteins are mutated [104,105]. Moreover, the upregulation of FOXO3a seems to be part of the protective cellular stress response by inducing autophagic activity in renal epithelial cells during hypoxia and subsequent renal injury [106,107]. Thus, tubular loss of FOXO3a induces the development of injury-induced chronic kidney disease due to reduced autophagic adaption in mice [107]. Previously, Zheng and colleagues demonstrated that FOXO3a is a substrate of PHD1-mediated hydroxylation [47]. An in vitro hydroxylation screening assay (CO_2_ capture assays) combined with recombinant PHD1 was used to detect novel PHD1 hydroxylation substrates. Using mass spectrometry, the authors could subsequently show that PHD1 hydroxylates FOXO3a at two proline residues (Pro426 and 437), which results in proteasomal degradation [47]. The regulation of FOXO3a by PHD1 is independent of HIF in this context and occurs at a post-transcriptional level since hydroxylation affected protein stability but not mRNA abundance [47]. Furthermore, immunoprecipitation and MS of DMOG-treated samples identified the recovery of USP9x deubiquitinase, which is known to stabilize proteins involved in cancer by preventing their degradation [108,109,110]. 

Notably, FOXO transcription factors repress Cyclin D1 expression, which plays an important role during cell proliferation [47,111]. This study speculates that the induction of FOXO3a by hypoxia might help conserve ATP by restricting cell proliferation as well as by reprogramming cell metabolism. In this regard, FOXO3a was reported to regulate oxygen consumption and reactive oxygen species (ROS) production by inhibiting mitochondrial gene expression [112,113]. Interestingly, a previous pre-clinical study found that loss of PHD1 and the subsequent up-regulation of PDK1 and 4 reduces the oxygen consumption of peripheral muscle cells by inducing the glycolytic flux, thus protecting cells from ROS-induced cell death during ischemia [114]. In this context, up-regulation of PDK4 alone via CRISPR/Cas9 gene-editing likewise protects hepatocytes from chemotherapy-associated ROS production and impairment of cell viability [115]. 

In conclusion, there is evidence to support FOXO3a as an alternative target for PHD1-mediated hydroxylation. Strategies to induce FOXO3a signaling by targeting PHD1 expression or PHI might have therapeutic potential, especially in chronic kidney diseases and in cancers such as breast cancer, in which the expression of FOXO3a or loss of Cyclin D1 inhibits cell proliferation [47,111]. 

### 6.4. MAPK6

Mitogen-activated protein kinase (MAPK) 6 is involved in angiogenic pathways, as MAPK6 controls the expression of VEGFR2, induces endothelial cell migration, proliferation and vascular tube formation [116]. In addition, several studies have highlighted the role of MAPK6 during cell differentiation [117,118,119]. 

Using a new pharmacological trapping strategy by treating cells with DMOG, combined with quantitative interaction proteomics, Rodriguez and colleagues provided evidence that both RIPK4 and MAPK6 are alternative targets for FIH- and PHD3-mediated hydroxylation, respectively [52]. Applying MS and immunoprecipitation they showed that PHD3 hydroxylates MAPK6 at proline 25, which potentially leads to its dissociation from the E3 ubiquitin protein ligase (HUWE1), thus protecting it from proteasomal degradation [52]. Conclusively, down-regulation of PHD3 activity with either PHI or siRNA reduced PHD3-conferred MAPK6 hydroxylation and decreased MAPK6 protein stability in vitro. 

### 6.5. Cep192

Several studies have demonstrated the crucial impact of hypoxia on the cell cycle by, for example, inducing G1/S1 cell cycle arrest both by HIF-dependent and HIF-independent mechanisms [120,121,122]. Central to cell division and thus the cell cycle, is the correct alignment of the chromosomes by the mitotic spindle, which is formed by centrosomes [123]. Centrosomes are also important for the formation of cilia, which are involved in cell sensing and movement [124].

A recent study demonstrated that PHD1 hydroxylates centrosomal protein (Cep) 192, which is a critical centrosome component, on the proline residue 1717 [53]. The authors first screened for proteins that contain the LXXLAP motif and are involved in spindle formation. Cep192, the most promising candidate, was shown to be colocalized and interact with PHD1 by immunofluorescence and immunoprecipitation, respectively, during mitosis [53]. Subsequent MS/MS confirmed PHD1-mediated hydroxylation of Cep192 on Pro1717 [53]. Further mutation experiments (proline 1717 to alanine) revealed that prolyl hydroxylation is crucial for Cep192 function as mutated cells showed a significant cell cycle arrest comparable to WT control cells depleted of Cep192 via genetic knockdown (siRNA) [53]. Additional immunoprecipitation results showed that hydroxylation of Cep192 by PHD1 seems to be important for Cep192 protein stability [53]. In summary, hydroxylation of Cep192 by PHD1 can be directly linked to cell cycle progression and cilia formation, thus explaining the oxygen sensitivity of these processes independent of HIF. 

### 6.6. ZHX2

Recently, it has been shown that the transcription factor, zinc fingers and homeoboxes 2 (ZHX2), functions as a tumor suppressor during the development of hepatocellular cancer and lymphoma [125,126]. By using a specific VHL-capture-based binding assay, Zhang and colleagues could screen a genome-wide human cDNA library, thus identifying ZHX2 as a new VHL substrate [54]. Inhibition of prolyl hydroxylation by DMOG or deferoxamine decreased ZHX2 binding to VHL and increased its protein expression in human kidney cells. MS detected proline 427, 440, and 464 as three ZHX2 prolyl hydroxylation sites, which could subsequently be verified by single mutations (P427A, P440A and P464A) and a mutant cell line that harbored all three mutations [54]. In tumor biopsies of patients with clear cell renal cell carcinoma (ccRCC) and confirmed VHL loss of function mutations, the authors could show a greater amount of ZHX2 (also of HIF-1α and -2α) in the majority of tumors compared to normal kidney tissue [54]. Subsequent functional in vitro and in vivo assays verified that ZHX2 promotes ccRCC carcinogenesis and NF-κB activity [54]. Notably, it remains unclear whether PHD1, 2 or 3 are conferring the hydroxylation of ZHX2. In summary, ZHX2 is a new hydroxylation-dependent VHL substrate which may thus represent an alternative therapeutic target for ccRCC. 

## 7. Hydroxylation of Non-HIF Targets Mediated by Factor-Inhibiting HIF (FIH) 

### 7.1. OTUB1

Ovarian tumor domain containing ubiquitin aldehyde binding protein 1 (OTUB1) is a deubiquitinase involved in cell metabolism [55,127] and is associated with poor prognosis in breast cancer patients treated with chemotherapy [128]. Overexpression of OTUB1 is associated with poor outcome and cancer progression in colorectal cancer [129,130]. Additionally, OTUB1 promotes tumorigenesis and cell invasion in prostate cancer [131].

Focusing on the metabolic aspects of OTUB1 signaling, Scholz and colleagues demonstrated, using MS and immunoprecipitation, that OTUB1 is hydroxylated by FIH on asparagine 22 (Asn22) which could be prevented by pharmacological PHD/FIH inhibition with DMOG and hypoxia [55]. DMOG-mediated impairment of Asn22 hydroxylation could be rescued by the overexpression of FIH. In cells expressing mutant OTUB1 (N22A), FIH activity measured by the turnover of 2-oxoglutarate into succinate (i.e., variant of the CO_2_ capture assay) was decreased compared to wildtype cells [55]. Notably, N22A mutation did not alter protein stability or deubiquitinase activity of OTUB1, but affected its interaction with other proteins especially those associated with metabolic processes [55]. Thus, resulting functional consequences could be observed, as cells overexpressing both FIH and N22A mutated OTUB1 (to minimize OTUB1 hydroxylation) demonstrated robustly increased phosphorylation of AMPKα under conditions of energy starvation [55]. This study thus provides new insights into the regulation of metabolism by hypoxia, which is at least, in parts, mediated through FIH-conferred Asn22 hydroxylation of OTUB1.

### 7.2. p105 and IκBα

As discussed above, NF-κB signaling is oxygen-dependent [132,133]. Both IκBα and one of its precursor proteins, p105, are characterized by the presence of ankyrin repeat domains (ARD) and function as NF-κB inhibitors [134]. Applying co-immunoprecipitation and substrate trapping via DMOG, it was demonstrated that FIH interacts with Asn678 of p105 [56]. Further analysis using a CO_2_ capture assay also revealed that p105 is a target for FIH-mediated hydroxylation. Of note, the CO_2_ capture assay detected several other putative FIH targets for hydroxylation including tankyrase-1 or gankyrin [56]. Sequence analysis of other ARD-containing proteins indicated comparable hydroxylation motifs in some family members (for example, IκBα, but not IκBβ) [56]. MS/MS confirmed two hydroxylation sites within IκBα: Asn244 and 210. Interestingly, Cockman and colleagues could detect a higher degree of hydroxylation of Asn244 (25–45%) than of Asn-210 (10–20%) [56]. Subsequent co-immunoprecipitation showed the physical interaction of FIH and IκBα [56]. Inhibiting FIH activity with hypoxia or DMOG prevented substrate hydroxylation. While a mutational change of both target residues (Asn to alanine) almost completely impaired IκBα-mediated inhibition of NF-κB DNA-binding activity, the study did not provide further direct evidence for biological consequences of IκBα hydroxylation on NF-κB activity [56]. However, it does add another dimension to the role of hydroxylation on the NF-κB pathway, potentially through FIH sequestration. 

Given the fact that FIH likely hydroxylates other ARD-containing proteins [67], that are involved in a broad spectrum of cellular processes, including cell adhesion, cell-cycle regulation or tumor suppression [135], the biological effects of FIH-dependent hydroxylation could potentially be broad.

### 7.3. RIPK4 

Receptor-interacting serine/threonine kinase (RIPK) 4 is known to interact with two isozymes of the protein kinase C family (PKCβ and PKCδ) [136,137]. Moreover, it has been shown that NF-κB signaling is, in part, regulated by RIPK4 [138] and that RIPK4 is involved in several oncogenic pathways, including RAF/MEK/ERK signaling and Wnt/beta-catenin signaling [139,140,141]. Multiple studies demonstrated that either up- [139,142] or down-regulation [143,144] of RIPK4 activity is associated with tumor progression and an unfavorable oncological outcome in several cancers. 

Rodriguez and colleagues demonstrated that RIPK4 is hydroxylated by FIH using a pharmacological trapping strategy with DMOG in combination with quantitative interaction proteomics [52]. FIH-mediated hydroxylation of RIPK4 did not affect protein stability. Interestingly, a T-cell factor/lymphoid enhancer-binding factor (TCF/LEF) luciferase reporter assay measuring β-Catenin activity showed that RIPK4 hydroxylation increased Wnt signaling pathway activity [52]. However, this study did not determine the exact hydroxylation site, as MS revealed four peptides all containing a hydroxylated asparagine residue [52]. 

## 8. Conclusions and Perspectives 

Non-HIF hydroxylation targets of PHDs and FIH are of great interest in terms of understanding the in vivo pharmacology and further clinical applications of PHIs, as they potentially increase the treatment spectrum of PHIs and could predict putative “off-target” effects of PHIs. 

By applying in silico genomic screening analysis (for example for the consensus sequence LXXLAP), protein association methods (for example immunoprecipitation) and pharmacological substrate-trapping with PHI followed by mass spectrometry analysis, increasing numbers of candidate targets for PHDs and FIH have been discovered, see Figure 4 and Table 1. Nonetheless, the screening, detection and verification of these targets remain a challenging task and recent studies showing protein hydroxylation by PHDs and FIH used a broad spectrum of different techniques to analyze and confirm these new substrates, see Table 1. Unlike the identification of phosphorylation, acetylation, and ubiquitination sites, detecting hydroxylation sites can be further complicated, as hydroxylation and oxidation can occur on multiple amino acid residues and side chains. Therefore, it is helpful to use pharmacological substrate trapping strategies to enrich substrate/enzyme interactions. Moreover, high resolution and coverage analysis such as tandem mass spectrometry in order to detect the correct site within a peptide should be applied. 

One disadvantage of screening techniques that detect enzymatic activity, such as the CO_2_ capture assay, is that the qualitative and quantitative amounts of enzymes and co-substrates are pre-determined by the experimental design and most likely do not always represent in vivo stoichiometry. The identification of alternative targets will, therefore, be limited by the experimental conditions under which the assay is performed. This explains, in part, conflicting results between the CO_2_ capture assays that are performed under different conditions (for example with or without cell lysate [25]). Yet unidentified components within the lysate could significantly influence highly controlled readouts, such as the CO_2_ capture assay. For this reason, future studies should apply a broad spectrum of different screening, detection and verification techniques to evaluate novel non-HIF targets of PHDs and FIH, see Table 1. We, therefore, propose that a combination of screening (for example, in silico or CO_2_ capture assay), detection (for example, MS/MS in combination with substrate trapping strategies) and verification (for example, immunoprecipitation in combination with cloning techniques) assays should be applied along with functional assays as a gold standard to identify novel functional non-HIF targets of PHDs and FIH, see Figure 3. 

The enzymatic reaction of PHD- and FIH-mediated HIF hydroxylation, see Figure 2, shows classical enzyme kinetics [145] that depend on pH, temperature, the concentration of the specific enzyme, co- and substrates and potential inhibitors [13,38,146]. Unlike for the hydroxylation of HIF this has not been shown for many of the new non-HIF substrates of PHDs and FIH. However, the increasing evidence for non-HIF targets of PHDs and FIH signifies that PHDs and FIH may have a broader specificity. 

Most studies to date have shown that the hydroxylation of proteins is directly associated with modified protein stability by either changing the interaction with protein deubiquitinases (for example, FOXO3a [47]) or ubiquitin ligases (for example MAPK6 [52]). Interestingly, as outlined above, hydroxylation can also alter the enzymatic activity of the specific proteins (for example, RIPK4 [52]) or change other post-translational modifications that, in turn, induce or repress their activity (for example, p53 [50]). In the case of OTUB1, hydroxylation even affects protein/protein interactions and binding patterns [55]. Notably, the question of whether alternative PHD hydroxylation is, in general, HIF-dependent or independent remains to be answered, as (for example, in the case of p53) there seems to be conflicting evidence showing that hydroxylation of p53 by PHD1 can be both HIF-dependent and independent [49,50]. Thus, it might be difficult to determine the exact physiological relevance of PHD- or FIH-mediated hydroxylation of non-HIF targets in some cases, especially, if the hydroxylation of these targets requires HIF. To further analyze the specific physiological relevance of hydroxylation of non-HIF targets, in vivo genetic knockout animal models that combine specific mutations of the known hydroxylation site within the investigated target and conditional HIF-1/-2 genetic ablations may be necessary. This might explain why, to date, only very few studies have investigated alternative non HIF-targets in vivo, see Table 1. 

In summary, protein hydroxylation by PHDs and FIH is neither unique nor ubiquitous but does affect a number of pathways, both HIF and non-HIF related. There is now mounting evidence that PHDs and FIH mediate protein hydroxylation of alternative targets other than HIF, see Figure 4 and Table 1. The technical improvement over the last few years and in the future will enable us to perform large-scale screening of additional substrates and targets for PHDs and FIH, thus allowing us to better understand how hydroxylation of these proteins influences cell signaling and pathophysiology. 

## Figures and Tables

**Figure 1 cells-08-00384-f001:**
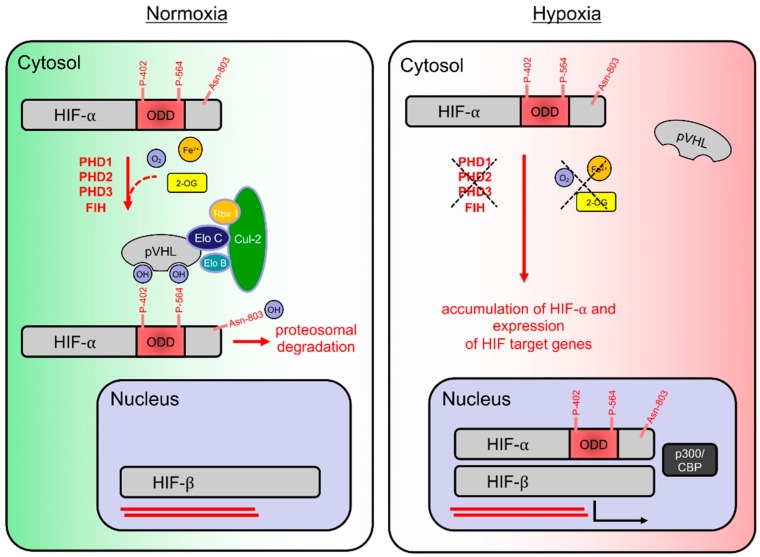
Schematic overview of the hypoxia-inducible pathway under normoxia (**left**) and hypoxia (**right**). Under normoxia, hypoxia-inducible factor (HIF) prolyl hydroxylases (PHD1–3) use oxygen (O_2_), iron (Fe^2+^), α-ketoglutarate (also known as 2-oxaloglutarate; (2-OG)) and ascorbic acid (vitamin C; not shown) as co-substrates to hydroxylate the HIF-1α subunits at two specific proline residues (human: Pro402 and 564) within the oxygen-dependent degradation domain (ODD), thus triggering recognition by the von Hippel-Lindau tumor suppressor protein (pVHL), which, as part of a E3 ubiquitin ligase complex (also containing elongin [Elo] C and B, cullin-2 [Cul-2] and RING-box protein [Rbx] 1), induces proteasomal degradation. HIF-1α hydroxylation by the factor-inhibiting HIF (FIH) at the asparagine residue (human: Asn803) additionally prevents binding of the transcriptional co-activator histone acetyltransferase p300/CREB-binding protein (p300/CBP) (left). In contrast, under hypoxia PHDs and FIH are unable to hydroxylate HIF-α subunits which accumulate and thus migrate to the nucleus, subsequently forming an HIF-complex with HIF-β subunits (right). Formed HIF-complexes bind to the DNA to induce numerous HIF target genes to counteract hypoxia (right).

**Figure 2 cells-08-00384-f002:**
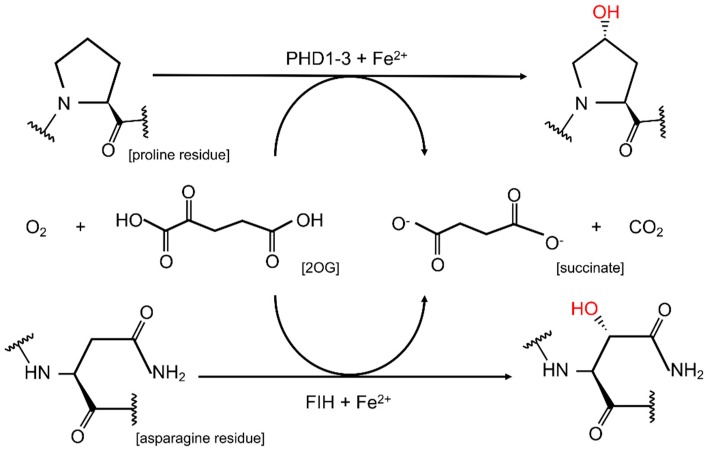
Enzymatic reaction of HIF-α subunits: HIF prolyl hydroxylases (PHD1-3) catalyze the hydroxylation of proline residues (**upper panel**) and factor inhibiting HIF (FIH) catalyzes the hydroxylation of asparagine residues (**lower panel**); both using oxygen (O_2_), iron (Fe^2+^), α-ketoglutarate (also known as 2-oxaloglutarate; 2-OG) and ascorbic acid (vitamin C; not shown) as co-substrates.

**Figure 3 cells-08-00384-f003:**
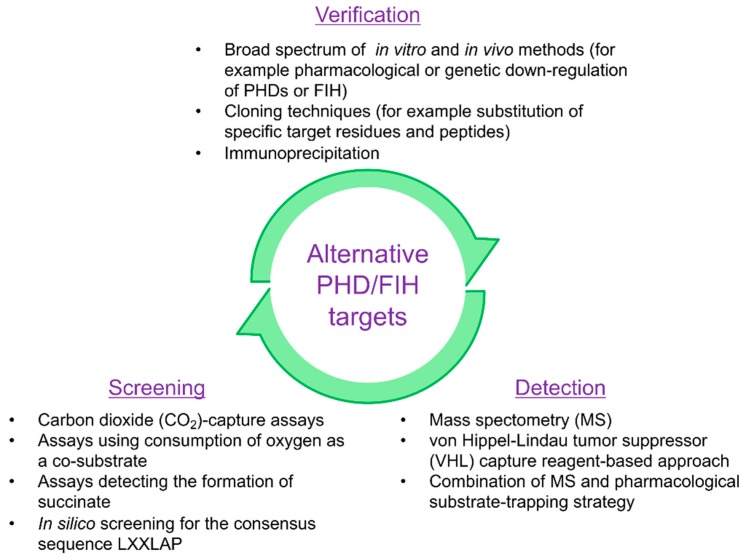
Overview of different techniques and methods to screen, detect and verify alternative targets and substrates of HIF prolyl hydroxylases (PHDs) and factor-inhibiting HIF (FIH).

**Figure 4 cells-08-00384-f004:**
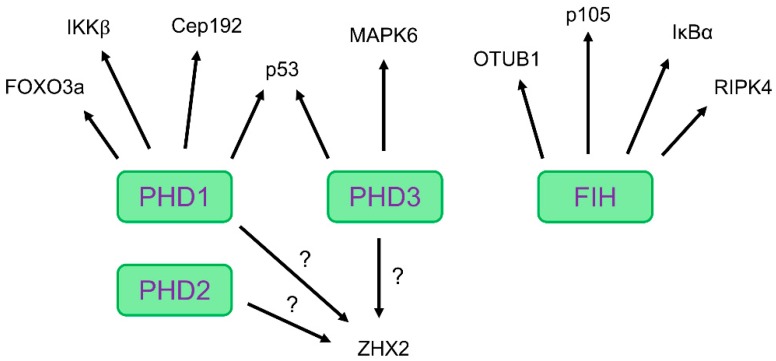
Schematic overview of alternative targets other than HIF mediated by HIF prolyl hydroxylases (PHD) 1, 3 or factor inhibiting HIF (FIH). PHD1-mediated hydroxylation: Forkhead box O3a (FOXO3a), IκB kinase-β (IKKβ) and centrosomal protein 192 (Cep192). PHD1- and 3-mediated hydroxylation: Tumor suppressor protein 53 (p53). PHD3-mediated hydroxylation: Mitogen-activated protein kinase 6 (MAPK6). FIH-mediated hydroxylation: Ovarian tumor domain containing ubiquitin aldehyde binding protein 1 (OTUB1), NF-κB precursor (NFKB1; p105), IκBα and receptor interacting serine/threonine kinase 4 (RIPK4). PHD-mediated hydroxylation: Zinc fingers and homeoboxes 2 (ZHX2).

**Table 1 cells-08-00384-t001:** Overview of applied methods to screen, detect and verify (alternative) hydroxylation targets of HIF-prolyl hydroxylases (PHDs) and factor-inhibiting HIF (FIH). Hydroxylation of HIF-1α by PHDs and FIH serves as a model in this case.

			Methods to Screen, Detect and Verify Hydroxylation						
Target	Enzyme [Residue]	Publication	CO2 Capture Assay and Other	In Silico and Other	MS, MS/MS and Other	Immunoprecipitation (e.g., GST Pulldown Assay)	VHL Capture Assay	Substrate- Trapping	Physiological Relevance *
HIF-1α	PHD1–3 [Pro402, 564]	[43,44,45]		✓	✓	✓	✓	Co, D or Fe	in vitro
HIF-1α	FIH [Asn803]	[46]	✓	✓	✓	✓		e.g., Co, Fe	in vitro
PHDs									
IKKβ	PHD1 [Pro191?]	[47,48]	✓	✓		✓		DMOG	in vitro
p53	PHD1 [Pro142]	[49,50]		✓	✓	✓		(DMOG)	in vitro, in vivo
p53	PHD3 [Pro359]	[51]			✓	✓		DMOG	in vitro
FOXO3a	PHD1 [Pro426, 437]	[47]	✓		✓	✓		DMOG	in vitro, in vivo
MAPK6	PHD3 [Pro25]	[52]		✓	✓	✓		DMOG/JNJ	in vitro
Cep192	PHD1 [Pro1717]	[53]		✓	✓	✓			in vitro
ZHX2	PHD? [Pro427, 440 and 464]	[54]			✓	✓	✓	e.g., DMOG	in vitro, in vivo
FIH									
OTUB1	FIH [Asn22]	[55]	✓	✓	✓	✓		DMOG	in vitro
p105	FIH [ASN678]	[56]	✓	✓	✓	✓		DMOG	in vitro
IκBα	FIH [Asn244 > 210]	[56]	✓	✓	✓	✓		DMOG	in vitro
RIPK4	FIH [Asn]	[52]		✓	✓	✓		DMOG	in vitro

Asn asparagine, Cep centrosomal protein, Co cobalt (II), *D* desferrioxamine, Fe Iron (II), DMOG dimethyloxaloylglycine, FOXO3a forkhead box O3a, GST glutathione S-transferase, HIF hypoxia-inducible factor, IKKβ IκB kinase-β, JNJ JNJ-42041935, MAPK6 mitogen-activated protein kinase 6, MS mass spectrometry, MS/MS tandem mass spectrometry, OTUB1 ovarian tumor domain containing ubiquitin aldehyde binding protein 1, PHD HIF-prolyl hydroxylases, p105 NF-κB precursor (NFKB1), p53 tumor suppressor protein 53, Pro proline, RIPK4 receptor interacting serine/threonine kinase 4, ZHX2 zinc fingers and homeoboxes 2. * Physiological relevance refers to the applied experimental design (in vitro–cell culture experiments versus in vivo–animal studies involved) upon first description.

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
