# Peer review of "Protein Hydroxylation by Hypoxia-Inducible Factor (HIF) Hydroxylases: Unique or Ubiquitous?"

_cells, 2019, doi:10.3390/cells8050384_

Round 1

Reviewer 1 Report

This is a very nice written review on the latest novel PHD and FIH substrates. This review paper will be very helpful for the people in the field. This review should also include the latest publication on identification of novel VHL substrate ZHX2, which will make this review more comprehensive. Also, there is a recent publication on FOXO3a (PMID: 30912765), hydroxylation and its role in kidney disease that can be discussed briefly.

Author Response

RESPONSE TO REVIEWER #1

This is a very nice written review on the latest novel PHD and FIH substrates. This review paper will be very helpful for the people in the field. This review should also include the latest publication on identification of novel VHL substrate ZHX2, which will make this review more comprehensive. Also, there is a recent publication on FOXO3a (PMID: 30912765), hydroxylation and its role in kidney disease that can be discussed briefly. 

General remark and reply:We thank the reviewer for her/his valuable and interesting comments. The reviewer raises some concerns, mainly regarding one recently identified VHL substrate, ZHX2 (Zhang et al., Science2018), which was not referred to in the previous version. Zinc fingers and homeoboxes 2 (ZHX2), functions as a tumor suppressor during the development of hepatocellular cancer and lymphoma (Yue et al., Gastroenterology2012 and Nagel et al., Leuk Res2012). Zhang and colleagues demonstrate that ZHX2 is hydroxylated at three proline residues (Zhang et al., Science2018). Subsequent functional in vitroand in vivoassays verified that ZHX2 promotes clear cell renal cell carcinoma (ccRCC) carcinogenesis and NFκB activity (Zhang et al., Science2018). Notably, it remains unclear whether PHD1, 2 or 3 are conferring the hydroxylation of ZHX2. Given the fact that ZHX2 could represent an alternative therapeutic target for ccRCC we added ZHX2 as a new target for PHD-mediated hydroxylation (see revised von Hippel-Lindau tumor suppressor (VHL) capture assay section, p. 9, ll. 11-13;new ZHX2 section, p. 17, l. 19 - p.18, l. 7;revised Table and new Figure 4). 

Furthermore, we included recently performed studies, which showed that the upregulation of FOXO3a seems to be part of the protective cellular stress response by inducing autophagic activity in renal epithelial cells during for example hypoxia and subsequent renal injury (Li et al., J Biol Chem2017 and Li et al., J Clin Invest2019). Thus, tubular loss of FOXO3a induces the development of injury-induced chronic kidney disease due to reduced autophagic adaption in mice (Li et al., J Clin Invest2019).We apologize that these recent publications were missing in the previous version of the review and thus thoroughly revised the FOXO3a section (see revised FOXO3a section, p. 15, ll. 9-13 and p. 16, l. 8).

In summary, we believe that the additional changes, which were made during this revision, significantly improved the quality of the present review.

Reviewer 2 Report

Strowitzki, Cummins and Taylor have put forward a review manuscript on a very timely and important topic; Is hydroxylation by the HIF hydroxylases unique or ubiquitous? They review the current methodology and the evidence of the recently reported non-HIFalfa substrates. There are certain issues the authors should still consider and modify the manuscript accordingly.

First, hydroxylation by PHDs and FIH is an enzymatic reaction that must obey the laws of enzymology; such as dependence on pH, temperature, substrate concentration and cosubstrates, and the effect of inhibitors. This is very seldom (if ever) shown for the non-HIFalfa substrates and should be discussed. Second, one of the key characteristics of enzymes is specificity towards their substrate. The emerging number of non-HIFalfa substrates would argue against high specificity. This should be discussed. Third, it would be helpful for the readers to indicate when DMOG is mentioned for the first time that it is a pan hydroxylase inhibitor, not a specific PHD or FIH inhibitor. Indeed, it is 4-25 times more potent inhibitor of collagen P4H than PHD1-3.

Lane 19: “Thus, PHDs are cellular oxygen sensors” is in some conflict with the previous sentence where the hydroxylation by both FIH and PHDs is said to decline in hypoxia. The PHDs are significantly more O2 sensitive than FIH.

Lane 23: The pharmacological inhibitors to treatment of anemia do not inhibit FIH.

Figure 1: Include FIH

Lane 71: should state: HIF prolyl 4-hydroxylases

Figure 2: include Fe2+ in the figure e.g. next to PHD1-3 and FIH

Lane 121: prolines in collagens are target for 4 and 3-hydroxylation carried out by two different enzymes. The collagen prolyl 3-hydroxylases should also be discussed here.

Lane 129: should state: …the Km values of PHDs of oxygen are higher than…

Table is very unclear to the reader and must be clarified. The “tabulators” should be omitted.

Page 7 lane 4: The title of the paragraph should be “Enzymatic assays or kinetic assays” and it should contain also the assay used to detect other enzymatic assays, such as (radioactive) 4-hydroxyproline developed originally for collagens by Juva K and Prockop DJ 1966, Anal Biochem 15, 77-83 and used e.g. in Koivunen P et al 2007 J Biol Chem 282, 30544-30552. Page 12 lane 257: mitotic spindle, which is ?INS?

Page 14 lane 362 hydroxylation even affectS

Author Response

RESPONSE TO REVIEWER #2

Strowitzki, Cummins and Taylor have put forward a review manuscript on a very timely and important topic; Is hydroxylation by the HIF hydroxylases unique or ubiquitous? They review the current methodology and the evidence of the recently reported non-HIFalfa substrates. There are certain issues the authors should still consider and modify the manuscript accordingly.

General remark:We thank the reviewer for the positive comments. The concerns raised during his/her revision mainly focused on the validity of alternative non-HIF targets of PHD and FIH regarding general principles of enzymatic reactions that should also rely to those newly identified targets. Other justified minor concerns uttered by this reviewer related more specifically to terminology and formatting of the figures. 

We believe that the additional changes made during this thorough revision substantially improved our present manuscript, and strengthen the case that the there is increasing evidence for alternative non-HIF targets of PHD and FIH. 

Reviewer #2 (major comment)

1. First, hydroxylation by PHDs and FIH is an enzymatic reaction that must obey the laws of enzymology; such as dependence on pH, temperature, substrate concentration and cosubstrates, and the effect of inhibitors. This is very seldom (if ever) shown for the non-HIFalfa substrates and should be discussed. 

2. Second, one of the key characteristics of enzymes is specificity towards their substrate. The emerging number of non-HIFalfa substrates would argue against high specificity. This should be discussed. 

Concerns #1 and 2 are replied together: Like every other enzymatic reactions the kinetics of PHD- and FIH-mediated hydroxylation is characterized by several general principles (Cornish-Bowden FEBS Lett 2013). The reviewer is correct in pointing out that this has been shown for the classical PHD and FIH target, HIF, but not for many of the recently newly identified alternative targets (Ivan et al., PNAS 2002 and Epstein et al., Cell 2001). 

Enzyme specificity towards substrates can range from “absolute specificity” (i.e. the enzyme will catalyze only one reaction) to “stereochemical specificity” (i.e. the enzyme will act on a particular steric or optical isomer). In this context, PHD and FIH seem to have a broad spectrum of specific substrates, which is in fact rather common to many enzymes, as they are capable of catalyzing secondary reactions (termed “promiscuous” reactions) in addition to the reactions that they have evolved to catalyse (Copley, Curr Opin Struct Biol 2017). Therefore, we included a new paragraph that discusses both of these facts within the revised conclusions and perspectives section, which reads as follows: 

The enzymatic reaction of PHD- and FIH-mediated HIF hydroxylation (Figure 2) shows classical enzyme kinetics [145]that depend on pH, temperature, the concentration of the specific enzyme, co- and substrates and potential inhibitors [38,146,147]. Unlike for the hydroxylation of HIF this has not been shown for many of the new non-HIF substrates of PHD and FIH. However, the increasing evidence for non-HIF targets of PHDs and FIH signifies that PHDs and FIH may have a broader specificity (seerevised conclusions and perspectives section, p. 21, ll. 20-25).

3. Third, it would be helpful for the readers to indicate when DMOG is mentioned for the first time that it is a pan hydroxylase inhibitor, not a specific PHD or FIH inhibitor. Indeed, it is 4-25 times more potent inhibitor of collagen P4H than PHD1-3.

The reviewer raises an important concern regarding the fact that Dimethyloxalylglycine (DMOG) is indeed a pan-hydroxylase inhibitor. Interestingly, DMOG is able to inhibit collagen prolyl hydroxylases, which play an important role during the formation of collagen fibres, more potently than HIF prolyl hydroxylases thus making them interesting therapeutics for fibrotic diseases (Hirsilä et al., J Biol Chem2003 and Manresa et al., Gastrointest Liver Phyisiol2016).

To better reflect this fact, we revised the section “Other prolyl hydroxylases” (seep. 6, ll. 9-12)and further introduced the term “pan hydroxylase inhibitor” upon the first appearance of “dimethyloxalylglycine (DMOG)” within the section “HIF-hydroxylation” as suggested by the reviewer (seep. 5, ll. 8f)

Reviewer #2 (minor comments)

1. Lane 19: “Thus, PHDs are cellular oxygen sensors” is in some conflict with the previous sentence where the hydroxylation by both FIH and PHDs is said to decline in hypoxia. The PHDs are significantly more O2 sensitive than FIH.

To address this concern we removed the sentence “Thus, PHDs are cellular oxygen sensors.” accordingly (seerevised abstract).

2. Lane 23: The pharmacological inhibitors to treatment of anemia do not inhibit FIH.

The reviewer correctly points out that the recently approved HIF prolyl hydroxylase inhibitor (PHI), Roxadustat, and other PHIs that are under clinical investigation for the treatment of anemia have little inhibitory effects on the factor inhibiting HIF (FIH) (Yeh et al., Chem Sci2017). We, therefore, corrected this sentence in the revised abstract, which now reads as follows: 

PHDs can be pharmacologically inhibited by a new class of drugs termed prolyl hydroxylase inhibitors which have recently been approved for the treatment of anemia due to chronic kidney disease (see revised abstract, p. 2, ll. 15ff).

3. Figure 1: Include FIH

We thank the reviewer for her/his comment regarding the previously missing factor inhibiting HIF (FIH) in Figure 1. Accordingly, we implemented FIH as suggested (see revised Figure 1).

4. Lane 71: should state: HIF prolyl 4-hydroxylases

We agree with the reviewer and corrected the sentence accordingly, which now reads as follows: 

Oxygen-dependent targeting of HIF-α subunits toproteasomal degradation is conferred by the HIF prolyl 4-hydroxylases PHD1 PHD2 and PHD3 (also termed EGLN2, EGLN1 and EGLN3 respectively) (see revised HIF-hydroxylation section, p. 3, ll. 126ff).

5. Figure 2: include Fe2+ in the figure e.g. next to PHD1-3 and FIH

We agree with the reviewer and thus included Fe2+as suggested in the revised version of Figure 2.

6. Lane 121: prolines in collagens are target for 4 and 3-hydroxylation carried out by two different enzymes. The collagen prolyl 3-hydroxylases should also be discussed here.

The reviewer is correct in pointing out that prolyl 4-hydroxylation and prolyl 3-hydroxylation are carried out by different enzyme families (reviewed in Hudson and Eyre, Connect Tissue Res 2013). To better reflect this fact we implemented a new paragraph within the section “other prolyl hydroxylases”, which reads now as follows: 

CPHs consist of two different enzyme families, collagen prolyl 4-hydroxylases and collagen prolyl 3-hydroxylases, which seem to have distinct functions [32]. Since prolyl 4-hydroxylation is, however, the single most prevalent post-translational modification in humans [33], we will focus on prolyl 4-hydroxylases within the present review (see revised other prolyl hydroxylases section, p. 5, ll. 21-24).

7. Lane 129: should state: …the Km values of PHDs of oxygen are higher than…

We thank the reviewer for his/her valuable remark and changed the sentence accordingly, which reads now as follows: 

Studies of enzyme kinetics revealed that the Michaelis constant(Km) values of PHDs for O2are higher than thoseof CPHs, which results in the capacity of PHDs to sense changes in cellular oxygen levels(see revised other prolyl hydroxylases section, p. 6, ll. 5ff).

8. Table is very unclear to the reader and must be clarified. The “tabulators” should be omitted.

We thank the reviewer for his/her remark and removed the table formatting (see revised Table).

9. Page 7 lane 4: The title of the paragraph should be “Enzymatic assays or kinetic assays” and it should contain also the assay used to detect other enzymatic assays, such as (radioactive) 4-hydroxyproline developed originally for collagens by Juva K and Prockop DJ 1966, Anal Biochem 15, 77-83 and used e.g. in Koivunen P et al 2007 J Biol Chem 282, 30544-30552. 

We agree with the reviewer that CO2capture assays belong to the spectrum of enzymatic or kinetic assays. We, thus, rephrased the sub-heading of the mentioned paragraph, which now reads as follows: 

Enzymatic or kinetic assays (seep. 7, l. 1). 

In addition, the reviewer correctly points out that there are several other enzymatic assays that rely on the detection of radioactively labelled substrates or products. We, thus, revised the new section “enzymatic or kinetic assays” and added a paragraph on the mentioned alternative assay, which unlike e.g. CO2capture assays directly determines the amount of radioactive 4-hydroxyproline formed during the enzymatic reaction and was initially developed for collagen (Juva and Prockop, Anal Biochem1966; Koivunen et al., J Biol Chem2007), which reads as follows:

Alternatively, the enzyme activity can be analyzed by directly determining the amount of radioactive 4-hydroxyproline formed within the substrate [49,50]One limitation that needs to be considered is that these kinetic assays are highly controlled in vitro studies, as purified enzymes and minimal co-substrates are typically used. (seerevised enzymatic or kinetic assays section, p. 7, ll. 12-16)

10. Page 12 lane 257: mitotic spindle, which is ?INS?

We changed the sentence accordingly, which reads now as follows: 

Central to cell division and thus the cell cycle is the correct alignment of the chromosomes by the mitotic spindle, which is formed by centrosomes[122].(see revised Cep192 section, p. 17, ll. 1f).

11. Page 14 lane 362 hydroxylation even affectS

We removed this typing error and changed the sentence accordingly, which reads now as follows: 

In the case of OTUB1, hydroxylation even affects protein/protein interactions and binding patterns(seerevised conclusions and perspectivessection, p. 21, ll. 3f).

Reviewer 3 Report

This is a well written and comprehensive review on an important topic. As pointed out by the authors PHD inhibitors are about to enter the clinics and it will therefore be of utmost interest to understand non-HIF mediated effects.

The review is well written and organized - I have only a few minor suggestions:

Fig 1 The nucleus under normoxia should contain HIF-1ß

Line 85 -it must read "...inhibits HIF-1a in normoxia "

Fig 2 Would you consider Vitamin C really as a co-substrate in the strictest biochemical sense?

Table still shows formatting

Line 145ff I believe to include the ref Winning et al J Immunol. 2010 Aug 1;185(3):1786-93 would be appropriate sine it shows PHD dependent regulation of NFkB effects

Lines 364/365 does not make sense

Conclusions: A cartoon showing the different non-HIF targets with their connections to PHD 1-3 and FIH might be good

incomplete refs #18, #22, #24

Author Response

RESPONSE TO REVIEWER #3

This is a well written and comprehensive review on an important topic. As pointed out by the authors PHD inhibitors are about to enter the clinics and it will therefore be of utmost interest to understand non-HIF mediated effects. The review is well written and organized - I have only a few minor suggestions.

General remark:We thank the reviewer for her/his valuable and intriguing comments. Overall the reviewer had several minor concerns regarding the outline and formatting of existing figures/tables and a missing reference that highlighted alterations in hydroxylase-dependent signaling with changes in NFκB activity. Moreover, we followed the valuable suggestion of the reviewer to include an additional figure that outlines the different non-HIF targets of PHDs and FIH.

We believe that the additional changes made throughout the manuscripts substantially improved our present review.

Reviewer #3 (minor comments)

1. Fig 1 The nucleus under normoxia should contain HIF-1ß

The reviewer correctly points out that the HIF-βsubunit is also present within the nucleus under normoxic conditions. This important fact was missing in the previous version of our Figure 1. We, thus, revised Figure 1 and included HIF-βin the nucleus under normoxia (see revised Figure 1, left scheme). 

2. Line 85 -it must read "...inhibits HIF-1a in normoxia "

We thank the reviewer for this valuable comment. Indeedfactor inhibiting HIF (FIH) inhibits HIF-α signaling in normoxia (not hypoxia) by hydroxylating an asparagine residue. We, thus, corrected this sentence, which now reads as follows:

Factor inhibiting HIF (FIH) is a second enzyme which inhibits HIF-α signaling in normoxia by hydroxylating an asparagine residue (Asn803 in human HIF-1α; Asn851 in human HIF-2α)[…] (see revised HIF-hydroxylation section, p. 4, ll. 14f).

3. Fig 2 Would you consider Vitamin C really as a co-substrate in the strictest biochemical sense?

The reviewer is concerned whether vitamin C (or L-Ascorbate) is indeed a co-substrate of HIF-prolyl hydroxylases (PHDs) in the strictest sense. We thank the reviewer for this interesting input. A co-substrate or -factor is indeed “any factor essentially required for enzyme activity or protein function” (Hashim and Adnan, Biochemical Education1994). In fact, L-Ascorbate is not strictly required for the in vitroreaction conferred by PHDs (Flashman et al., Biochem J2010). 

Since L-Ascorbate nonetheless enhances the activity of PHDs (Jaakkola et al., Science2001 and Hirsilä et al., J Biol Chem2003), we believe that vitamin C should be defined as a co-substrate of PHD-mediated enzymatic reaction. We, therefore, did not change the labelling of vitamin C as a co-substrate throughout the manuscript and the figure legends and hope that this is acceptable for the reviewer. However, we are obviously willing to change the labelling of vitamin C if the reviewer should request us to do so.

4. Table still shows formatting

We thank the reviewer for his/her remark and removed the table formatting (see revised Table).

5. Line 145ff I believe to include the ref Winning et al J Immunol. 2010 Aug 1;185(3):1786-93 would be appropriate sine it shows PHD dependent regulation of NFkB effects

We thank the reviewer for this remark regarding the study of Winning and colleagues showing the prolyl hydroxylase-dependent but HIF-independent activation of hypoxia-induced monocyte-endothelial adhesion under acute hypoxic conditions (Winning et al., J Immunol2010), which overall supports alterations in hydroxylase-dependent signaling with changes in NFκB activity. We apologize that this work was missing in the previous version of the manuscript and included the reference within the revised version of the manuscript (see revised NFκB section, p. 12, ll. 15f).

6. Lines 364/365 does not make sense

The reviewer is correct that the above-mentioned paragraph was difficult to understand for the reader. We, therefore, re-phrased this sentence to better reflect our notion that indeed the question whether alternative PHD hydroxylation is in general HIF-dependent or independent remains to be answered, as (for example in the case of p53) there seems to be conflicting evidence showing that hydroxylation of p53 by PHD1 can be both HIF-dependent and independent (see revised conclusions and perspectives section, p. 22, ll. 4-7).

7. Conclusions: A cartoon showing the different non-HIF targets with their connections to PHD 1-3 and FIH might be good

The reviewer rightly points out that a cartoon highlighting the different non-HIF targets of PHDs and FIH might improve the quality of the present review. We shar the opinion of the reviewer and thus included a new figure 4 (see new Figure 4). 

8. incomplete refs #18, #22, #24

We thank the reviewer for his/her comment regarding the incomplete references of #18, 22 and 24. We implemented the appropriate complete references accordingly (see revised reference section, p. 28f).